# Methylcellulose–Cellulose Nanocrystal Composites for Optomechanically Tunable Hydrogels and Fibers

**DOI:** 10.3390/ma14185137

**Published:** 2021-09-07

**Authors:** Ville Hynninen, Jani Patrakka

**Affiliations:** 1Faculty of Engineering and Natural Sciences, Tampere University, P.O. Box 541, FI-33720 Tampere, Finland; jani.patrakka@tuni.fi; 2Department of Applied Physics, Aalto University, P.O. Box 15100, FI-00076 Espoo, Finland

**Keywords:** methylcellulose, cellulose nanocrystal, hydrogel, birefringence, wet-spinning, optical fiber, thermoresponsive, LCST, nanocomposite

## Abstract

Chemical modification of cellulose offers routes for structurally and functionally diverse biopolymer derivatives for numerous industrial applications. Among cellulose derivatives, cellulose ethers have found extensive use, such as emulsifiers, in food industries and biotechnology. Methylcellulose, one of the simplest cellulose derivatives, has been utilized for biomedical, construction materials and cell culture applications. Its improved water solubility, thermoresponsive gelation, and the ability to act as a matrix for various dopants also offer routes for cellulose-based functional materials. There has been a renewed interest in understanding the structural, mechanical, and optical properties of methylcellulose and its composites. This review focuses on the recent development in optically and mechanically tunable hydrogels derived from methylcellulose and methylcellulose–cellulose nanocrystal composites. We further discuss the application of the gels for preparing highly ductile and strong fibers. Finally, the emerging application of methylcellulose-based fibers as optical fibers and their application potentials are discussed.

## 1. Introduction

Cellulose, in general, is a renewable and almost inexhaustibly available natural polymer that can be processed into various forms, ranging from pulp to microfibers to nanoparticles, as well as to chemically derivatized polymer chains [1]. Each cellulose derivative displays its own characteristics and functionalities depending on its processing history, for instance: high surface charge, water solubility, lyotropic liquid crystallinity, thermoresponsivity, and extreme mechanical strength. Consequently, a range of fully cellulose-derived advanced and functional materials and (nano)composites can be prepared by selecting the correct raw materials and their combinations [2,3,4,5,6]. Furthermore, the inherent relative ease of modifiability makes cellulose applicable in a wide range of conditions and enables synergies with other materials, such as carbon nanotubes, graphene, other biomolecules, and metallic nanoparticles [7,8,9,10,11,12]. Potential applications for advanced cellulosic materials include: biomedicine, tissue engineering, photonics, smart materials, emulsions, foaming, and sensors [13,14,15,16,17]. The rich history and applications of cellulose and various cellulose-based materials and nanocomposites are evident from several excellent reviews [4,5,18,19,20,21,22,23,24]. Yet, there are numerous properties and potential applications of celluloses, nanocelluloses, and their composites that remain unexplored. Especially, the trend towards sustainably produced optical and optoelectronic materials and devices has recently gained attraction, and cellulose derivatives and their composites appear as key components [20].

This review focuses on the recent findings on the optomechanical properties of methylcellulose (MC), a commonly used cellulose derivative, and its composites with cellulose nanocrystals (CNCs). The review is divided into six sections and a conclusion. The first section provides an overview of methylcellulose hydrogels and cellulose nanocrystals and their physicochemical properties. Importantly, aqueous MC solutions display thermosensitive structural changes at the molecular level that result in fully reversible mechanical and optical properties. CNCs, on the other hand, are rod-like nanoparticles known for their excellent stiffness and lyotropic liquid crystalline behavior. Therefore, CNCs offer a mechanical reinforcement in composite systems. A typical procedure to prepare MC-based nanocomposite hydrogels is also covered. The second and third sections cover the mechanical tunability of the MC-CNC composite hydrogels. The base level of the gel stiffness and flow behaviors can be significantly enhanced and controlled by altering the amount of CNC dopants. Furthermore, the inherent thermoresponsivity allows for a fully reversible control to stiffen further and “relax” the gels at will. The fourth section covers the optical characteristics of the composite gels. Their transparency and color are significantly controlled by the composition, dopants, and surrounding temperature. Additionally, at suitable conditions, birefringence may be induced and reversibly enhanced by temperature, which suggests tunable and dynamic structural order within the hydrogels of certain compositions. In the final two sections, we discuss the preparation of MC-based fibers from the MC-CNC hydrogels by simple wet-spinning in benign conditions. Furthermore, their potential as optical fibers is discussed. MC-based fibers, generally, show a high ductility and good functionalization potential, for example, with luminescent gold nanoclusters. The optical clarity and structural smoothness of the fibers make them prospective biopolymer optical fibers. In general, cellulosic optical fibers are a relatively new and little-explored field, where MC fibers have been reported among the first successes. Overall, this review demonstrates the high versatility and significant application potential available even in some of the simplest cellulosic materials, specifically focusing on the MC-based nanocomposites.

## 2. Methylcellulose and Cellulose Nanocrystals

MC is the simplest chemical derivative of cellulose, where the available hydroxyl (-OH) groups on the cellulose polymer backbone are partially substituted with methoxyl (-OCH_3_) groups (Figure 1a) [25]. A suitable degree of substitution (typically 1.2–2.0) imbues MC with water solubility and fully reversible lower critical solution temperature (LCST) behavior, setting it apart from the native cellulose (Figure 1b–d) [26]. However, the molecular weight, concentration, degree of substitution, and substitution pattern along the MC chain affect the exact gelation conditions and characteristics [26,27,28,29]. Nonetheless, aqueous MC solutions typically convert into turbid gels when heated above ca. 40 °C and regain their transparency and flow properties upon cooling [30]. Upon the heating-induced gelation, the MC polymer chains transition from random coils to fibrillar aggregates, as schematically represented in Figure 1c,d), resulting in macroscopic mechanical stiffening and increased optical scattering.

MC has attracted significant commercial attention in various fields, driven by exceptional thermosensitive gelation properties, rheological characteristics, non-toxicity, biocompatibility, and relative structural simplicity. Its applications range from general viscosity modifying agents to food technology and biotechnology, among others [25]. MC provides a naturally functional and physically crosslinked water-based matrix or scaffolds within nanocomposites, whose stiffness is feasibly tunable from viscous fluid to stiff self-standing gels by simply adjusting the MC concentration or the surrounding temperature [32,33]. The charge-neutral and amphiphilic nature of the MC also ensures a good compatibility and the homogeneous mixing with a wide range of polymers, dopants and modifying agents, without sacrificing the characteristically inverse thermoreversible behavior [31,34,35,36]. Accordingly, the versatility of the MC hydrogels is highlighted and essential for the MC-CNC nanocomposite hydrogels and fibers.

CNCs, on the other hand, represent a completely different type of cellulosic material; they are particularly known for their high mechanical strength, rigidness, and colloidal rod-like appearance [37,38,39]. Hence, CNCs are often considered as potential strengthening agents and reinforcing nanofillers, which also applies to the MC-CNC nanocomposites [40]. Importantly, the surface functionality, and thereby the properties, of CNCs can be modified depending on the preparation methods, their source and post-chemical modifications [41,42,43]. Additionally, the CNCs self-assemble into left-handedly twisted chiral nematic (i.e., cholesteric) liquid crystalline structures above a critical concentration (c*), which may serve as a scaffold for hierarchical structural materials, to increase the mechanical order and strength in composites, and to adjust the optical properties of a material, among other functions [44,45,46,47]. Both the mechanical and liquid crystalline properties are relevant for the MC-CNC nanocomposites.

When combined into nanocomposite hydrogels and fibers, MC and CNCs express multiple complementary properties and synergistic effects: mechanical adjustability, thermally tunable stiffness and structural stability, feasible spinnability, tunable transparency, and controllable structural degradability, which are all achievable through relatively straightforward and gentle processing [31,48,49]. Thus, versatile and extraordinary materials from freely flowing liquids to stiff and solid nanocomposites can be prepared from relatively simple cellulose-derived components. This also emphasizes the potential of cellulosic materials in general. Next, the special characteristics of MC and CNCs are presented in more detail; followed by the discussion of the corresponding nanocomposite hydrogels; the manufacturing of such hydrogels into highly ductile fibers; and, lastly, their transformation into cellulose-derived optical fibers, and the potential of these fibers.

### 2.1. Preparation and General Properties of MC Hydrogels

Hydrogels are viscoelastic materials consisting of either chemically or physically crosslinked three-dimensional (3D) networks encapsulating a large number of water molecules. Gels, in general, are states of matter intermediate between solids and liquids containing at least two components, viz., a continuous phase (i.e., a solid network) and dispersion media (typically a liquid or solvent) [50]. Therein, solutes and water molecules may flow rather freely within the gel matrix, while a stable structure and shape are, nonetheless, maintained. Despite containing a large amount of liquid, the gels display solid-like mechanical properties under oscillatory rheological experiments. Typically, strong gels show an order-of-magnitude higher storage modulus (G′) than a loss modulus (G″) [51]. The gel’s rigidity and mechanical properties can be adjusted and enhanced via the addition of rigid supportive nanofillers, such as high-aspect-ratio CNCs [40,48].

When MC is dissolved in water, the polymer chains readily hydrate, resulting in a transparent viscous fluid (Figure 1b). However, proper measures are encouraged to ensure homogeneous MC dissolution and to avoid undesired aggregation during the preparation phase. Due to the LCST behavior, the desired amount of dry MC powder is advisably first added into hot water (>80 °C) to avoid its untimely and uneven hydration, as recommended by the commercial MC manufacturers [25,49]. Because of its LCST behavior, the dry MC can be evenly mixed and distributed MCs into hot water by stirring without immediate gelation and significantly increased viscosity. The mixture remains liquid-like provided that a high temperature is maintained. The hydration and closer interactions with water molecules are triggered only after the MC powder is homogeneously distributed and surrounded by the hot solvent. This is done by cooling down the system below the LCST temperature to ensure an even dissolution. Continuous stirring is recommended during the gelation of the wetted MC powder until it has been properly hydrated and gelated to prevent the sedimentation of the non-gelated MC and gradient formation. The solution turns from cloudy to transparent upon the cooling-induced MC hydration as the polymer chains become hydrated [28]. When preparing nanocomposite hydrogels, water-soluble nanoparticles may be feasibly mixed with the dry MC powder in hot water prior to the initial gelation to facilitate the homogeneous distribution along with the MC polymer.

Dilute solutions of MC typically show liquid-like behavior at low temperatures (below the LCST temperature) and flow readily [25]. Higher viscosities and more gel-like properties can be reached by increasing the concentration or molecular weight of the MC, or via the addition of rigidifying (nano)components [27,33,48,52]. Shear thinning, which facilitates fiber-spinning methods involving narrow spinnerets or capillaries, is characteristically observed under steady shear experiments at low temperatures, while the transformation to shear thickening may be observed close to or above the gelation temperature [53]. This behavior can be further controlled by substituting the MC polymer chains with chemically cross-linkable side groups, such as allyls, which allow solid, stable materials regardless of the temperature [54].

At elevated temperatures above the LCST gelation point, the storage and loss moduli of MC solutions increase considerably, and turbid hydrogels (G′ >> G″) are achieved (Figure 1b–d and Figure 2) [33]. However, in addition to the gel composition, the exact gelation temperature is affected by the heating rate [28]. A significant change in volume or syneresis, (e.g., the excretion of network-bound water) does not occur upon the MC gelation, unless heating is continued for substantially extended periods of time, such as days or weeks [55]. However, considerable hysteresis is observed upon cooling back to the solution state, even though the thermally induced gelation is otherwise fully reversible [28,33,56]. The hysteresis has been suggested to be induced by the slow unraveling and low re-hydration rate of the fibrillar MC bundles present in the gel stage [28]. As indirect evidence, the addition of negatively charged highly hydrophilic CNCs significantly reduces the observable hysteresis by facilitating the gel re-hydration upon cooling (Figure 2) [48].

When aqueous MC is heated, the MC polymer chains assemble into a percolative network consisting of fibrillar structures with a high water content of ca. 60% and a rather constant and persistent diameter of ~15 nm, which has been confirmed and visualized both experimentally (cryo-TEM imaging, and small angle neutron and X-ray scattering) and via computational simulations (Figure 2) [57,58,59,60,61,62,63]. Additionally, fibril formation may be induced by the addition of salt [64,65]. Rather recently, the tightly packed core region of the fibers has been suggested to comprise semicrystalline twisted MC chain domains aligned along the fibril long axis and surrounded by amorphous and less-ordered regions [66]. The percolation of the semi-stiff fibril network contributes to the increased macroscopic gel stiffness and turbidity at elevated temperatures. The current knowledge of such thermally induced MC fibrillar aggregates and their significance on the MC gelation process has been comprehensively summarized in the recent reviews by Morozova and Coughlin et al. [67,68].

Furthermore, the MC fibrils may grant the strain stiffening property to the hydrogels, which is relatively rarely encountered in carbohydrate-based hydrogels [69]. Instead, strain stiffening is a characteristic feature of various extracellular matrix components and cytoskeletons, such as collagen, fibrin, actin, and vimentin [70,71]. In strain-stiffening systems, the material’s stiffness increasingly grows as a response to increasing deformation, to resist over-extension and to prevent catastrophic mechanical failure [70]. In MC hydrogels at elevated temperatures, strain stiffening has been reported in samples consisting of MC chains with a relatively high molecular weight (300,000), and in MC-CNC nanocomposite hydrogels containing MC chains with a lower molecular weight (88,000). In the MC-CNC composites, the strain stiffening was rendered detectable by the addition of CNCs, suggesting an intriguing synergistic interplay between the MC network and the nanodopants [31,53].

Finally, aqueous solutions of MC chains with a high molecular weight (≥380,000) have been shown to exhibit concentration, molecular weight, and temperature-dependent liquid crystallinity [72]. Thus, in addition to ordered fibrillar structures upon gelation, MC appears to have an inherent tendency for collectively structured assemblies. By adding suitable nanodopants favoring liquid crystalline packing, such as CNCs, the order can be further effectively enhanced [73]. Overall, MC has been shown to readily interact with CNCs both in aqueous conditions and as stabilizers for oil-water emulsions and double-morphology latexes [73,74,75]. Furthermore, MC’s ability to efficiently interact at the interfaces between hydrophilic and hydrophobic substances can allow surprisingly rigid and controllable structures as demonstrated with the stabilization of oil-in-water emulsions [76]. In these emulsions, the droplet coalescence and the stabilization ability and structure of the surrounding MC layer are significantly adjusted by simply tuning the oil/solid fat composition of the oil droplets.

### 2.2. Characteristics of Cellulose Nanocrystals

Cellulose nanocrystals (CNCs) are highly crystalline rod-like high-aspect-ratio nanoparticles that are obtained via top-down deconstruction from cellulosic raw materials (Figure 3) [18]. Depending on the source and processing conditions, the length of the CNCs is typically hundreds of nanometers and their aspect ratio ranges from ~5 to over 100 [77]. For instance, cotton-based CNCs typically measure 70–300 nm in length and around 7–10 nm in lateral dimension (Figure 3) [77]. Often sulfuric acid hydrolysis is preferred since it produces CNCs with a high negative surface charge (emerging from sulfate half ester groups), a high hydrophilicity, and an excellent colloidal stability [78,79]. During the processing, the more loosely packed and amorphous regions become degraded, while the organized crystallites remain as the CNC particles. The benefits and drawbacks of various preparation methods have recently been thoroughly reviewed by Vanderfleet and Cranston [18]. Currently, the industrial scale production of CNCs is readily available, making the material suitable for large scale manufacturing and applications [80].

CNCs are structurally very strong and stiff, which makes them excellent reinforcing units for nanocomposites. As such, the mechanical properties of individual CNCs supersede, for example, steel or glass fibers, and are comparable to Kevlar [81,82]. However, due to their anisotropic rod-like crystalline packing, the mechanical performance of CNCs is also direction-dependent, and the longitudinal elastic modulus of the CNCs (~150 GPa) far exceeds the transverse modulus (~8–57 GPa) [37,83]. In order to maximally transfer the beneficial properties of CNCs to nanocomposites, it is important to ensure homogeneous distribution, alignment and interactions of the nanorods and to prevent particle aggregation. The CNC surface typically contains a multitude of hydroxyl (-OH) groups that allow hydrogen bonding and efficient coupling to the surrounding matrix to provide reinforcement [41]. These groups, along with electrostatic interactions, may also be used to chemically adjust the CNC functionalities or to drive certain interactions and self-assembled structures [7,12,84,85,86]. However, at certain conditions, CNCs may self-aggregate, which results in the loss of available surface groups and surface area, and in the potential formation of structural defects, weak points, and local inhomogeneities, which potentially compromise the mechanical benefits of the nanocomposites [87]. Thus, ensuring proper distribution and stability during material processing is essential.

CNCs display lyotropic liquid crystalline behavior, that is, they spontaneously form liquid crystalline assemblies when dispersed in water at sufficiently high concentrations [88,89]. For example, the typical critical concentration (c*) required to trigger the onset of the phase separation of aqueous cotton-based CNCs into ordered liquid crystalline and disordered phase is approximately 3–4% [88]. While individual CNCs are structurally right-handedly twisted, collectively, they tend to form a left-handed chiral nematic, i.e., cholesteric, liquid crystalline assemblies (Figure 3c) [7,44,90]. The chirality can be retained of upon drying and used for templating hierarchical solids [47,91]. In nanocomposite hydrogels and solids, the organized assemblies may serve as an additional source of structural stability and as a template for anisotropic mechanical and optical properties, such as oriented swelling, birefringence, and structural colors [14,46,77,92,93,94,95]. However, achieving the c* and feasible CNC packing while being embedded inside a polymer matrix or a hydrogel may pose some challenges. On the other hand, the polymer network may also facilitate the packing via particle confinement or co-alignment [31]. Additionally, in polymer nanocomposites, the matrix polymer has been shown to form a semi-solid interface on top of the nanodopants, thus, contributing their effective size and volume fraction, and simultaneously lowering the actual c* and potential percolation threshold [96,97].

## 3. Mechanical Tunability of MC-CNC Composite Hydrogels

While pure aqueous MC solutions appear transparent at room temperature (ca. 22 °C), both the mechanical and optical properties can be significantly adjusted by adding nanodopants. In general, water-soluble nanoparticles are feasibly mixed within the MC matrix during the gel preparation process [31,36,48,52]. First, the addition of high-aspect ratio stiff nanorods, such as CNCs or chitin nanocrystals (ChNC), effectively induces the sol–gel transition and also enables self-standing MC-based hydrogels to form at low temperatures [31,48,52]. It has already been shown that an addition of ~1.0% of sulfuric acid hydrolyzed cotton-based CNCs (length ~200 nm) into 1.0% aqueous MC (MW 88,000), that is, with a total solid content of 2.0%, results in the gelation with the storage modulus (G′) > loss modulus (G″), and G′ > 10 Pa at room temperature [31,48]. Thus, a proper gel is achieved, even though the nanocomposite structure comprises of 98.0% water. Furthermore, within the identical 1.0% MC solutions, G′ has been observed to exceed G″ by adding as low as 0.5% of cotton-based CNCs. However, the composite structure remains very delicate and weak, with the absolute G’ of only ca. 5–8 Pa [31,48]. For comparison, a pure aqueous MC solution of identical molecular weight with a higher total solid content of 4.0%, still remains a flowing liquid with a G″ above the G′, regardless of its significantly higher absolute G’ of ca. 170 Pa in the linear viscoelastic range [36].

The nanocomposite gel’s stiffness can be increased further by increasing the concentration of CNCs within the MC matrix (Figure 4). However, the mechanical enhancement does not proceed linearly as a function of the CNC amount, but shows a rapid initial increase followed by a more plateau-like region, suggesting a percolation phenomenon [31]. Generally, the effect of CNC content has been studied in MC-CNC nanocomposite gels, where the MC content has been kept constant (1.0%) while varying the amount of CNCs [31,36,48]. Hence, the gel’s total solid content and the relative MC-to-CNC ratio are modified simultaneously. A similar approach, accompanied by MC composition and a tunable nanodopant composition, is used for exploring MC-ChNC nanocomposite hydrogels [52]. Experimentally, the addition of cotton-based CNCs up to ~1.5% is reported to enhance the gel stiffness, above which the reinforcement efficacy decreases [31]. It has been suggested that above the 1.5% CNC loading, the inter-CNC interactions begin to outweigh the particle–matrix interactions. This results in the less effective entanglement and reduced reinforcement capacity [31]. In total, by tuning the CNC content from 0 to 3.5% while keeping MC constant at 1.0%, approximately a two orders-of-magnitude increase in G’ from ~1 Pa to ~100 Pa is achievable at room temperature (RT) (Figure 4) [31,48]. Simultaneously, with the onset of the plateau-like region at around 1.5% of added CNC, birefringence observable with a polarized optical microscope appears, which is discussed below, implying a percolation threshold [31].

In the literature, only the use of cotton-based CNCs have been reported with MC gels, but the required CNC content to enable the initial gelation and significant reinforcement is expected to diminish when utilizing CNCs with higher lengths and aspect ratios, such as those extracted from tunicates (length ~1000 nm) [98]. Higher-aspect-ratio particles are known to display better reinforcement potential and to lower the percolation threshold of nanocomposites in general [96,99]. Likewise, the MC network consisting of higher MW polymer chains can provide better entanglement and increased gel stiffness. On the other hand, small metal nanoclusters mixed into MC hydrogel are observed to promote more liquid-like behavior, soften the gel network, and reduce G’ values [36]. This is suggested to be caused by the tiny size (*d* ~3 nm), spherical shape, and surface structure of the metal clusters that do not provide sufficient additional rigidity or stiff entanglement points for the MC polymer chains like the CNCs. Thus, the MC–metal cluster interactions are unable to provide reinforcement and tend to loosen the structure [36]. The observation strongly agrees with the nanocomposite literature, where, in general, fine nano objects with high aspect ratios and prominent surface-to-volume ratios are known to facilitate both favorable particle–matrix and particle–particle interactions, and to provide superior reinforcement compared to simpler spherical nanodopants [100]. Nonetheless, the possibility of using tiny nanodopants to tune the MC rheological properties towards a more liquid-like state may also open interesting opportunities for processing and material properties in the future. Additionally, nanodopants may simultaneously provide additional benefits and properties, for instance, sensing and photoluminescence, to the nanocomposites.

The thermoreversible response characteristic of pure MC is retained in the MC-CNC nanocomposite hydrogels regardless of the CNC addition, which allows for the reversible stiffening of the structure upon heating (Figure 4). That is, the baseline stiffness and sol–gel characteristics of an MC-CNC, or an MC gel modified with any analogous nanoparticles, at low temperatures can be adjusted by controlling the amount of nanodopants and gel composition. Furthermore, the fully reversible additional stiffening of approximately one order-of-magnitude may be triggered on demand by increasing the temperature (Figure 4b,c). Hence, a two-step and reversible method to adjust the MC-CNC or any analogous MC-nanodopant hydrogel exists. Moreover, the typical hysteresis of MC hydrogels upon cooling can be reduced by the inclusion of hydrophilic CNCs or nanodopants, which facilitate the rehydration of the entire nanocomposite network [48].

## 4. Optical Birefringence in MC-CNC Composite Hydrogels

The emergence of birefringence in MC-CNC hydrogels has been observed at certain CNC loadings, specifically at and above ~1.5% of cotton-based CNCs when keeping the amount of MC (MW 88,000) constant at 1% (Figure 5) [31]. The appearance of birefringence matches with the onset of diminishing mechanical returns from the addition of CNCs as mentioned above and suggests percolation. Birefringence indicates two important features, viz., (i) the aligned structural order exists and can be maintained within the composite hydrogel, and (ii) the polarization and direction of the incoming light dictates how it will refract through the sample material. Birefringence can be observed by using polarized optical microscopy (POM), which is typically used to visualize and identify liquid crystalline phases and materials [101]. Thus, structural order and stability can be imparted onto MC gels by adding CNCs. However, it should be noted that the total observed intensity of birefringence may be enhanced by the local shear induced alignment emerging from the contact between the material and the used imaging chamber walls [31]. Nonetheless, the structure is also able to maintain the possible shear-determined orientations. Thus, opening possibilities for the manufacturing of nematically oriented composite materials. For instance, anisotropic acrylamide hydrogel-bound CNC films have already shown prospects, such as ionic strength and pressure sensors [102]. The addition of CNCs also affects the macroscopic light scattering and gel transparency by turning the general appearance from clear (with pure MC) to an increasingly grayish and cloudy color, with the increased CNC content [31,36,48]. As a rule of thumb, nanoparticles exceeding the size limit of ca. 50 nm can be expected to significantly increase the light scattering, which is then observed as enhanced opaqueness [103].

Birefringence is characteristic of cholesteric liquid crystalline assemblies, and typically such behavior emerges in pure high-concentration CNC (above 3–4%) dispersions. It is atypical to encounter liquid crystalline assemblies at such a low CNC content of ~1.5% (a total hydrogel solid content of 2.5%) when used in the MC-CNC composites. Importantly, this value is below the usual c* of 3–4% for the cotton-based CNCs [31,88]. Therefore, the MC matrix appears to contribute to CNC alignment and confinement. Furthermore, even though liquid crystallinity has been reported in some pure high-molecular weight MC solutions, MC solutions with a lower MW (88,000) have shown no birefringence in similar conditions. However, it could be a transformed birefringent by the addition of CNCs [31,72]. Hence, MC-induced CNC confinement and the mutual alignment of MC chains in the MC-CNC nanocomposite hydrogels are suggested to result in the increased structural order above the percolation threshold. Consequently, it manifests as birefringence and is significantly tunable by the CNC content [31]. Potentially, interactions between the carbohydrate backbones of MC and CNCs provide additional support and rigidity to the composite hydrogel, similar to mixtures of cellulose nanofibers (CNFs) and chitosan in composite fibers [104].

When the CNC concentration exceeds the percolation threshold, the particle–particle interactions eventually begin to outweigh the particle–MC matrix interactions. Consequently, gel rigidness increases, and local CNC-driven and more organized regions emerge which are, then, observed as birefringence [31]. A similar concentration-dependent aggregation has been observed in solid fiber nanocomposites, as is discussed further below. It is possible that gel preparation conditions can affect the orientation and possible interactions of the CNCs. For example, fast cooling and gelation may lock the particles more randomly, while slow and gentle cooling could allow time for more controlled CNC rod assemblies within the emerging gel matrix. By carefully controlling the processing conditions it is observed that the liquid crystalline capabilities of the CNCs within the MC nanocomposites can be better utilized to create, for example, internally hierarchical, mold-cast nanocomposites.

Like the mechanical response, birefringence in MC-CNC gels also responds to changes in temperature; above the gel transition temperature (ca. 40 °C), the birefringence reversibly increases (Figure 5). While the magnitude is relatively modest compared to G′, the alterations, nonetheless, suggest a tunable packing and structural order [31]. It has been hypothesized that, upon heating, the fibrils formed by MC mutually co-align and pack themselves with the CNCs which leads to the observed reversible increase of birefringence [31]. MC fibrils may also increase the local confinement of the CNCs and reduce the freedom of movement. Upon cooling, the MC chains relax back towards random coil formation, which loosens CNC packing and allows more mobility, dims the birefringence and causes a loss of structure [31].

In summary, the baseline mechanical and optical properties of the MC-based nanocomposite hydrogels are readily tunable. Importantly, the effect of the concentration of MC and the composition of reinforcing nanoparticles, especially the cotton-based CNCs, have been described here in more detail [31,48]. In general, high-aspect-ratio particles enable good mechanical reinforcements and birefringent structures at surprisingly low concentrations. The interaction between the gel matrix and the nanoparticles is deemed necessary in determining the structural strength, packing, and birefringent properties, which is highlighted especially at the elevated temperature, where the thermosensitivity of MC chains facilitates both mechanical and optical response enhancement. Furthermore, the characteristics of the added nanocomponents contribute to the gel clarity, color, and potential additional functionalities, and thus, also broaden the application potential. For instance, CNC-derived birefringent patterns are suggested to be safe and forgery-proof security markers [105].

## 5. MC-CNC Composite Fibers

Methylcellulose solutions and MC-based nanocomposite gels at sufficient compositions can be extruded into fibers via wet-spinning, a process that has previously been applied to CNF hydrogels [49,106]. The solution, an MC-based hydrogel with a desired composition and potentially modified with dopants, is pumped through a thin capillary tube into a coagulation bath, where the extruded fiber solidifies (Figure 6) [106]. The capillary tube forces the gel into a fibrillar shape and exerts some shear alignment onto the material flowing through it. A pump, such as a syringe pump, guarantees the replicable and stable flow of the material. The process is performed at ambient temperature (ca. 22 °C) without the need for chemical crosslinkers. Typically, ethanol that is mildly polar and water-miscible is used as the coagulation anti-solvent. Ethanol replaces water in the MC structure and promotes interactions between the MC polymer chains, resulting in solid fibers. Finally, the fibers can be lifted from the coagulation bath and, after the drying and evaporation of ethanol, solid macroscopic fibers are obtained [36,49,106]. Since no covalent crosslinkers are used, the randomly oriented MC chains may rearrange and stretch within the fibers as a response to stress which makes them particularly ductile. As with hydrogels, adding dopants or reinforcing agents can be used to tune the processing and fiber properties. In general, high-aspect-ratio CNCs improve the wet strength, rigidity, and facilitate the handling of the nascent fibers, while, for instance, tiny gold nanoclusters are observed to soften the spinning dope and increase fiber softness similar to hydrogels [36,49].

When spinning pure MC, a high molecular weight and sufficient concentration facilitate polymer chain entanglement and fiber strength and, thus, determine the optimal spinning conditions. Instead, MCs of a low molecular weight and at low concentrations fail to produce fibers and freely diffuse into the coagulation bath [49]. Naturally, the operating limits of the extrusion setup need to be taken into consideration. The polymer alignment, fiber size, and fiber mechanical properties can be controlled by adjusting the extrusion capillary diameter, capillary length, and flow speed [106]. In general, a high flow speed and shear stress tend to improve fiber stiffness and internal alignment when spinning cellulosic fibers. However, the ductility and ultimate strain often become sacrificed in the process [106,107,108]. As a broadly applicable post-modification, stretching may be used to further enhance fiber alignment and mechanical strength [107].

The addition of CNCs generally reinforces the solid composite fiber structure, similar to hydrogels. However, optimal properties are typically achieved at lower-end CNC loadings; with 1% or less CNC loadings in the MC-based hydrogel spinning dopes, where the CNCs provide entanglement points and structural rigidity. At the same time, this allows the MC chains to flow and reorganize under stress, confirming the typical CNC-reinforced polymer nanocomposite behavior [49,109,110]. Examples of the undesired effects of high CNC dopant concentration and their aggregation, and consequential macroscopic structural changes are reported in MC-CNC, alginate-CNC, and in poly(vinyl alcohol) (PVA)-CNC fibers, among others [49,110,111,112]. Typically, after exceeding the optimal CNC concentration, the particles tend to aggregate and significantly stiffen the fiber. This results in the overall sub-optimal mechanical performance, including high brittleness, and inhomogeneous structure. At such conditions, the rearrangement of the CNCs towards cholesteric or twisting assemblies, which are characteristic of pure CNCs, may emerge. Therefore, the microphase separation of liquid crystalline regions or the larger-scale structural twisting and buckling of the fiber structure are observed (Figure 6) [49,110,111,112]. On the other hand, if homogeneous CNC distribution is successfully maintained, the structural reinforcement effect analogous to the ordered β-sheet units within the natural silk fibers may be achieved [113].

Specifically, pure MC, and MC-CNC fibers with a low CNC loading (typically ~20% or less of CNCs from the total solid content of fibers) coagulate in high concentration ethanol, show a good transparency, and have a relatively smooth morphology (Figure 6) [49]. However, due to gravitational pull and competing inward- and outward-bound solvent diffusion processes during the coagulation phase, fibers with fully spherical cross-sections are challenging to obtain (Figure 6) [114]. Nonetheless, the aligned homogeneous arrays of rod-like building nanoscale blocks that conform to the fiber shape can be observed within fractured fiber cross-sections (Figure 6) [36,49]. The currently reported MC-based fibers have also shown high ductility up to 30–50% depending on the composition, but a relatively low Young’s modulus (<8 GPa) and ultimate strength (<200 MPa) (Figure 7) [36,49]. Compared to the literature, the magnitude of ductility of the MC-based fibers has rarely been achieved with other fully cellulose-derived fibers, which typically break within a 10% strain [49]. However, significantly higher moduli and maximum stress have been reported; highly aligned CNF-based fibers prepared by Mittal et al. have reached an elastic modulus of 86 GPa and maximum stress of 1.57 GPa [49,115]. The strength and stiffness could likely be further enhanced to a certain extent by optimizing the spinning conditions. However, so far, research has mainly focused on exploring the effects and characteristic features of different MC-CNC fiber compositions and concentrations, rather than the optimization of technical processing parameters [36,49]. Nonetheless, the optimal composition of the MC-CNC fibers for mechanical performance (that is, minimizing the CNC content) produces the highest transparency and relatively smooth and homogeneous morphology [36]. These properties are beneficial for optical fiber capabilities and potential applications that are discussed in the next section.

Since the MC-based fibers are prepared through a simple coagulation treatment without covalent chemical crosslinkers, they also show interesting degradability in wet conditions that is tunable by the surrounding temperature due to the thermosensitivity of the underlying MC matrix [36]. The MC matrix allows for the rapid intake of water and the swelling of the fiber, which is potentially beneficial for sensing applications, but it also results in full fiber degradation in approximately 4 h if immersed in pure water at room temperature (ca. 22 °C) [36]. This is a potential drawback when considering, for example, biological applications involving tissues and cell culture conditions with high humidity. However, due to the LCST behavior of MC, MC fibers are shown to stabilize at a biologically relevant temperature of 37 °C and survive for extended periods of time (at least 6 h without the loss of structure) in extreme humidity in a water bath. Therefore, the fiber stability and the lifespan of the MC-based fibers are feasibly extended by relatively moderate heating to 37 °C [36].

## 6. MC-Based Biopolymeric Optical Fibers

Optical fibers are an essential part of modern data and communications networks, and they hold several benefits over traditional copper wire-based electronics. Optical fibers, in general, have substantial information capacity, lower signal loss and more affordable material costs. They also generate less heat, and are lightweight and insensitive to electromagnetic disturbances and energy spikes [116]. The current commercial optical fibers are typically made from silica glass or plastics, such as poly(methyl methacrylate) (PMMA) [116]. The glass optical fibers (GOFs) are optimal for long-range communication networks, with a very low signal attenuation of approximately 0.2 dB/km [116]. That is, the signal can be transmitted across tens of kilometers without irreparable signal degradation or the need for amplification. Plastic optical fibers (POFs), on the other hand, typically express approximately one order-of-magnitude higher attenuation coefficients and a thicker multimode profile, which makes them more suitable for local networks and applications [116].

However, both GOFs and POFs possess certain limitations or suboptimal features for specific occasions that could be potentially overcome or reduced by utilizing cellulosic, or more generally, biopolymeric raw materials [117]. For instance, while GOFs are superior in long-range communication, they are stiff, potentially brittle, nondegradable, may be challenging to modify or dope, and require extreme temperatures to produce and frequent special handling skills, in order to preserve their performance across connections and necessary junctions. POFs partially fix some of the problems as they can be feasibly produced via extrusion, are thicker, softer, more lightweight, modifiable and cost-effective [116]. However, the standard POFs cannot avoid issues in degradability and sustainability, even though they tend to react to the environment and may become more easily damaged by temperature and humidity than the GOFs [116].

Thus, preparing optical fibers from biopolymeric materials, particularly from cellulose, with high environmental sensitivity, ease of modifiability, and degradability, is suggested to complement and support the traditional POF technologies in cases where the current properties are not optimal [36,118]. That is, the main intention of biopolymeric and cellulose-derived optical fibers is not to directly compete with the GOFs and POFs, but to complement them where appropriate properties and functionalities are needed. One significant reason for this is that the current biopolymeric optical fibers are effectively usable only at distances on a submeter scale; however, this application is useful for biological or biomedical devices and sensors [119,120,121,122,123]. Nonetheless, cellulose, generally has the potential to allow novel, feasibly modifiable, and multifunctional optical fibers for short-range applications and devices from renewable and nearly endless raw materials. The preparation processes of cellulosic fibers are benign and mild compared to the traditional GOFs, whose preparation requires temperatures of up to 2000 °C [124].

So far, a limited number of attempts to prepare fully cellulose-derived optical fibers, including the MC-based optical fibers, have been reported [36,118,125]. Nonetheless, the potential of such fibers beyond simple light transmission is already envisioned since they have demonstrated the potential for use in drug delivery and sensory applications, controlled degradation, and feasibly modifiable functional fiber optic nanocomposites [36,118,125]. On the other hand, a range of cellulosic components have been frequently coupled with traditional optical fibers and into photonic devices to utilize, for instance, their sensitivity to humidity or the structural colors and chirality of liquid crystalline CNCs which broaden the application potential and cooperation with traditional light transmission methods [13,126,127,128]. Overall, work still remains, especially in finding and optimizing the best cellulose derivatives for optical fiber preparation for different purposes and standardizing such fibers’ quality and processing conditions before any larger scale production becomes topical. To date, most of the research focusing on cellulose fiber spinning, in general, has strongly focused on the optimization of the mechanical properties [129]. While it is a valuable approach in itself and potentially largely translatable into the manufacturing of the cellulose-derived optical fibers, it is unfortunately still rather limited and neglects aspects, such as transparency, that are essential for optical fibers and requires more active research.

At present, biopolymeric optical fibers, either in solid or gel form, have been reported from silk, agarose, and alginate, among others [119,120,121,122,123,130,131,132]. In addition, even collections of a bacterial organism have been successfully utilized to modulate light propagation analogous to fiber optics, despite being used for significantly shorter length scales [133,134]. Nonetheless, such bacterial experiments may turn out critical in providing inspiration and insights into how to efficiently control and tune light propagation inside biopolymeric materials and composites. The state-of-the-art biopolymeric optical fibers can reach a signal attenuation in the range of 0.2–1.0 dB/cm, which is corroborated by the reported cellulose-derived optical fibers [119]. Out of the explored biomaterials, spider silk is logically of high interest because of its excellent mechanical properties and high transparency [123,135,136]. However, the availability of spider silk may turn out as the limiting factor when considering large-scale production, even though synthetic biology potentially holds the keys to solve the issue [137]. In comparison, wood-derived optical fibers from a near endless renewable raw material, and with a long research history aiming towards good mechanical performance, appear as a tempting alternative. Moreover, cellulose-based hydrogels are widely established and generally non-toxic, and thus, could also allow cellulose-derived hydrogel fibers [138]. Hydrogel fibers are interesting especially for tissue and biomedical applications where mechanical tunability (to match the properties of living tissue), sufficient porosity, cell encapsulation potential, diffusion of nutrients, and biocompatibility are essential [120,139].

Specifically, pure methylcellulose-derived optical fibers have reached the best attenuation coefficient (α) of 1.47 dB/cm, measured with cutback methods, which is on par with various reported silk-based optical fibers (Figure 8) [36,119,122,123,132]. In addition, it is similar to the other two up-to-date reported cellulose-derived optical fibers consisting of cellulose acetate-coated regenerated cellulose, and hydroxypropyl-filled cellulose butyrate tubes whose attenuations remain in the range of ~1.0–5.9 dB/cm [118,125]. Furthermore, it is important to notice that the reported MC-based fibers do not have any cladding, and in the future, by adding a suitable coating to the structure the scattering can be potentially lowered further [36]. However, as a drawback, the antisolvent-dependent coagulation and processing of the MC fibers may not be optimal for upscaling due to its extended coagulation period compared to, for example, the highly efficient fiber spinning by ionic interactions [36,49]. In the future, combinations and the use of different cellulose derivatives have the potential to tune both the mechanical and optical capabilities of the fibers, since various cellulose derivatives allow a multitude of potential combinations and fabrication routes for optical fibers.

Of the reported MC optical fibers, the ones with the thinnest cross-sections generally show the lowest signal attenuation [36]. Since the fiber thickness is partially affected by the initial hydrogel spinning dope concentration, its tunability is closely related to the mechanical properties; low concentrations typically allow softer, more ductile, and thinner fibers with lower attenuations. The higher concentrations produce thicker and stiffer fibers with relatively high signal loss [36,49]. This is logical as the thicker optical fibers allow the transportation of a higher number of simultaneous propagation modes, resulting in an overall increase in signal dispersion [116].

The mechanical strength can also be increased via the addition of CNCs. However, nanoadditives often tend to contribute to higher signal loss due to increased light absorption and scattering due to their highly absorbant chemical groups, particle sizes matching to visible light wavelength, and the increased amorphous crystalline boundary regions within the material [36,103]. Thus, the feasibility of adding nanodopants to the MC fibers to achieve complementary functions often needs to be balanced against the loss of transparency.

The effective operating range due to the relatively high signal attenuation of >~1.5 dB/cm of the MC-based optical fibers is limited to a centimeter scale even without any modifications and, thus, the fibers are only usable for short-distance applications and sensors [36]. Therefore, as is acknowledged from the start, the aim of the MC-based optical fibers is not to pursue superior long-distance performance and usurp the GOFs; the additional loss caused by the dopants may often be acceptable and not compromise the envisioned applications. Moreover, the dopants may allow multifunctional optical fibers by coupling the material with sensory or luminescent properties, or higher tunable mechanical strength, as has been demonstrated via the addition of gold nanoclusters (AuNCs) (Figure 8d,e) [36,140,141,142]. While such properties and analogous dopants can be typically added to GOFs mainly via their modified cladding layers, in MC fibers, and other biopolymeric fibers, it is possible to make the actual fiber core active and responsive to its environment while still allowing for the further finetuning of the properties via suitable coatings [36,118,120,126,130]. The combination of functional cores and claddings appears as a fruitful ground for further research. For instance, double-core cellulose-derived optical fibers from cellulose butyrate and hydroxypropyl cellulose have already been envisioned as candidates for drug delivery and sensing purposes, in addition to the typical signal transmissions [125].

## 7. Conclusions

Methylcellulose is one of the extensively utilized cellulose derivatives. Despite its interesting phase-changing behavior and thermoreversibility, several of its properties have only been identified recently. More importantly, the structural transition and self-assembly that drives the inverse thermoresponsive behavior was only understood in recent findings using advanced cryo-TEM imaging analysis. The compatibility of methylcellulose with structurally and functionally diverse nanoparticles allow access to various composite systems. As discussed in this review, the addition of one-dimensional cellulose nanocrystals reinforces the gels and solid composites. The shear thinning properties of the gels offer a facile and environmentally benign route for wet spinning without the need for chemical cross-linking. Rather surprisingly, the MC and MC-CNC composite fibers display high ductility and strength that is on par with other biopolymeric fibers prepared using highly sophisticated methods and technologies. More importantly, the high optical transparency of MC-based fibers offers a new opportunity to explore their application in waveguiding. The MC fibers are shown to act as short-distance optical fibers with a low attenuation coefficient, encouraging more active research towards cellulosic optical fibers. Furthermore, several nanoadditives can be incorporated into the MC matrix to enhance their optical properties and functionalities, allowing, for example, luminescent fibers or the sensing of toxic metal ions [36].

## Figures and Tables

**Figure 1 materials-14-05137-f001:**
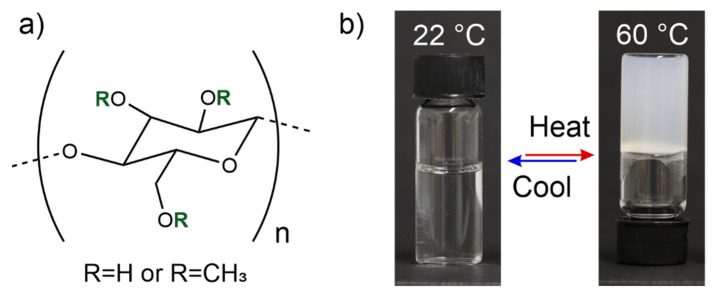
Methylcellulose. (**a**) Chemical structure of methylcellulose repeat unit. (**b**) Photographs of 1% MC aqueous solution at 22 °C and 60 °C. Due to the LCST behavior, MC turns into a turbid gel at elevated temperatures. (**c**,**d**) Schematics of aqueous MC polymer chain (blue line corresponds to MC polymer chain) behavior at 22° and 60°, respectively. The blue spheres in (**c**,**d**) correspond to the methylcellulose anhydroglucose unit, i.e., components of the polymer chain, whose structural formula is presented in (**a**). Figure (**c**,**d**) is adapted with permission from [31], (https://pubs.acs.org/doi/full/10.1021/acs.biomac.8b00392, date accessed on 3 September 2021). Copyright 2018 American Chemical Society.

**Figure 2 materials-14-05137-f002:**
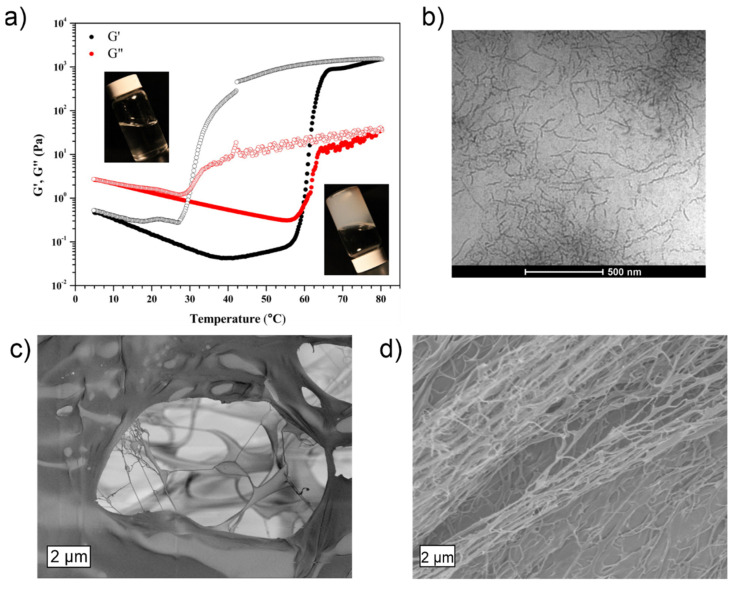
Thermoresponsive behavior of methylcellulose aqueous solution. (**a**) Development of storage (G′; black) and loss (G″; red) moduli of 1.4 wt% MC (molecular weight, MW 300,000) upon a heating (solid circles) and cooling (open circles) cycle. Insets show photographs of the sample at 25 °C and 75 °C. (**b**) Cryo-TEM image of 0.20 wt% MC solution (MW 300,000) above the LCST threshold reveals the fibrillar MC aggregates that form at elevated temperatures and facilitate gelation. (**c**) SEM micrograph of a 1% MC hydrogel freeze-dried from room temperature (ca. 22 °C) and then fractured. (**d**) SEM of a 1% MC hydrogel freeze-dried and fractured from 75 °C shows the enhanced presence of fibrillar structures. (**a**,**b**) adapted with permission from [57]. Copyright 2013 American Chemical Society. (**c**,**d**) adapted with permission from [31]. Copyright 2018 American Chemical Society.

**Figure 3 materials-14-05137-f003:**
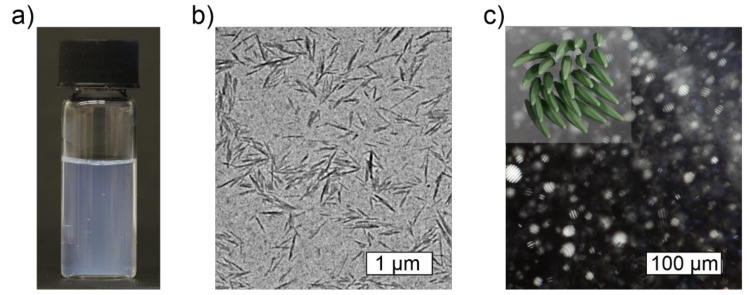
Cellulose nanocrystals. (**a**) A photograph of 1 wt% aqueous solution of cotton-based CNCs produced using sulfuric acid hydrolysis, i.e., below the critical liquid crystalline concentration. It appears grayish due to scattering and behaves like water. (**b**) TEM image of cotton-based CNCs. (**c**) Polarized optical microscopy image showing cholesteric tactoids in an 8.0% cotton-based CNC solution, i.e., above the critical liquid crystalline concentration. The inset shows a schematic illustration of a left-handed cholesteric CNC assembly.

**Figure 4 materials-14-05137-f004:**
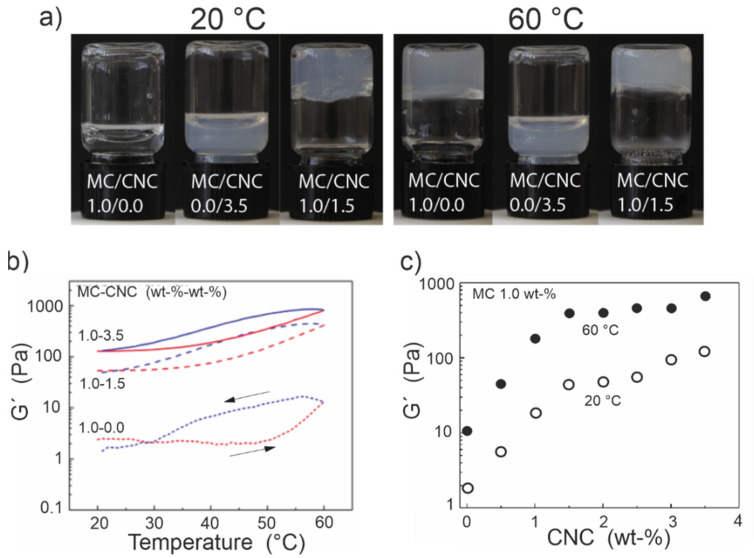
Mechanical properties of MC-CNC nanocomposite gels. (**a**) General appearance of pure MC, pure CNC, and a MC-CNC mixture compared at 20 °C and 60 °C. (**b**) Changes of storage moduli (G′) for three different MC-CNC nanocomposite hydrogels with heating and cooling cycle. The CNC content tunes the baseline stiffness when MC is kept constant, but all gels show fully reversible and thermally induced stiffening. (**c**) Storage modulus (G′) of MC-CNC hydrogels as a function of the CNC loading, when MC content is fixed, at 1% at 20 °C and 60 °C. Heating allows further reversible stiffening of ca. one order of magnitude. Note the logarithmic Y-axis scale. Adapted with permission from [31] (https://pubs.acs.org/doi/full/10.1021/acs.biomac.8b00392, date accessed on 3 September 2021). Copyright 2018 American Chemical Society.

**Figure 5 materials-14-05137-f005:**
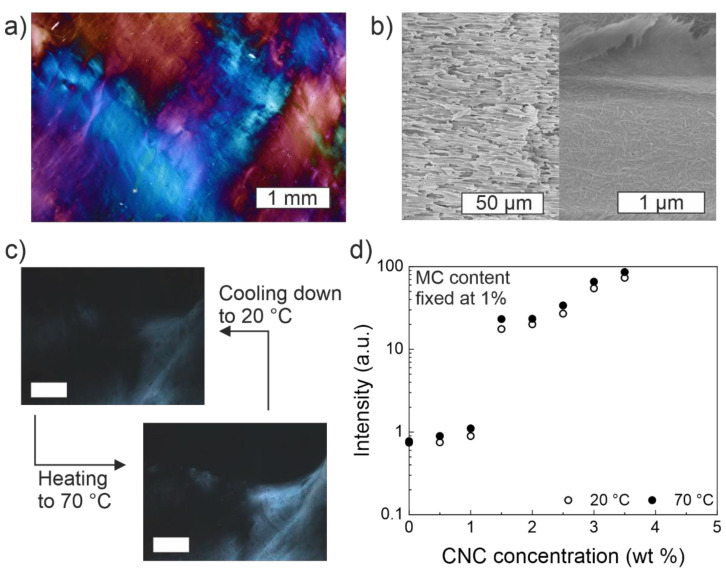
Birefringence in MC-CNC composite hydrogels. (**a**) MC-CNC 1–2.5% hydrogel imaged through crossed polarizers at 20 °C. A full-wave plate is used for enhanced colors. (**b**) SEM micrographs of a MC-CNC hydrogel freeze-dried and fractured from 70 °C at two different magnifications show CNCs dispersed in the MC medium. (**c**) The birefringence of MC-CNC gels intensifies upon heating to 70 °C and recovers upon cooling down to 20 °C. (**d**) Increase of birefringence intensity as a function of the hydrogels CNC content when MC is kept fixed at 1.0%. Note the logarithmic scale to highlight the differences also at the lower CNC loadings. Birefringence is visible above the intensity threshold of ~1. Adapted with permission from [31] (https://pubs.acs.org/doi/full/10.1021/acs.biomac.8b00392, date accessed on 3 September 2021). Copyright 2018 American Chemical Society.

**Figure 6 materials-14-05137-f006:**
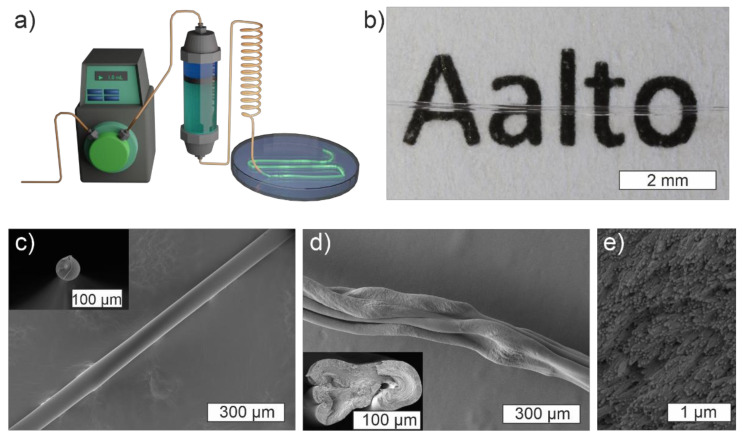
Fiber spinning and morphology. (**a**) Schematics drawing of wet-spinning setup components: a pump, sample container, capillary tube, and a coagulation bath. (**b**) Transparent MC fibers prepared from 3% MC hydrogel. (**c**) SEM image of a fiber wet-spun from 2% MC hydrogel showing smooth and homogeneous morphology. The inset shows the cross-sectional shape of the respective fiber. (**d**) SEM micrograph of fiber spun from a MC-CNC 1.0–3.0% nanocomposite hydrogel, showing buckled and twisted shape. The inset shows an uneven cross-sectional shape. (**e**) Fractured cross-section of a fiber wet-spun from a 4.0% MC hydrogel shows an array of aligned internal nanoscale structures. (**d**) Adapted with permission from [49], copyright 2019 Elsevier; and (**a**–**c**) and (**e**) adapted with permission from [36], copyright 2021 John Wiley and Sons.

**Figure 7 materials-14-05137-f007:**
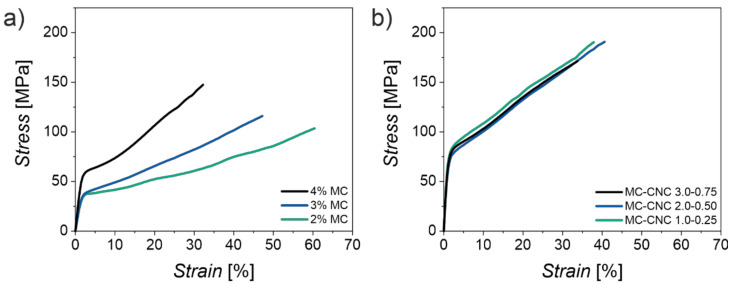
Mechanical properties of the MC-based fibers. (**a**) Stress–strain curves of pure MC fibers wet-spun from 2%, 3%, and 4% MC hydrogels (MW 88,000). Increased total solid content of gel appears to stiffen the resulting fibers. With low solid content, exceptional ductility above 50% is achieved. (**b**) Stress–strain curves of MC-CNC fibers with different total solid contents but with identical MC–CNC ratios (4:1). There the difference in total solid content of composite hydrogel is reflected mostly in the fiber diameter, whereas significant alterations in stiffness or ductility are not observed. Adapted with permission from [36]. Copyright 2021 John Wiley and Sons.

**Figure 8 materials-14-05137-f008:**
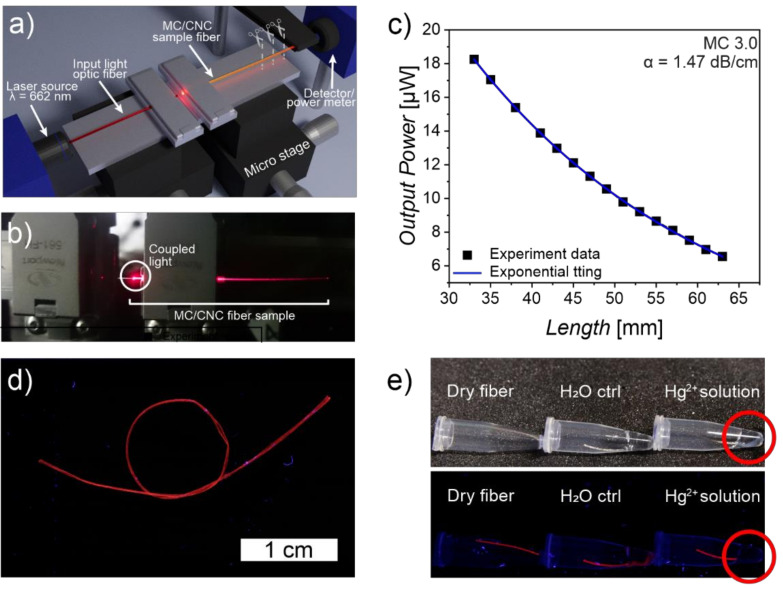
MC-based multifunctional optical fibers. (**a**) The attenuation of MC-based fibers has been studied with the cutback method, where the transmitted light intensity has been measured, the fiber shortened, and the measurement cycle repeated for several rounds. The results reveal signal attenuation per length of unit. (**b**) Photograph of an attenuation measurement system with an MC fiber coupled to a light source. (**c**) Representative attenuation measurement data (black circles) and mathematical fitting to the data (blue line). (**d**) MC fiber doped with gold nanoclusters (AuNC) is photoluminescent and shines red light upon UV irradiation. (**e**) AuNC-modified MC fiber as Hg^2+^ sensor. Under UV light, the absorption of Hg^2+^ ions (10 mM solution) into the MC-AuNC fiber quenches the luminescence providing a visual cue of the presence of Hg^2+^. The luminescence of a dry MC-AuNC fiber and a similar control fiber immersed in pure water remains unaltered. Adapted with permission from [36]. Copyright 2021 John Wiley and Sons.

## Data Availability

Data sharing is not applicable for this article.

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
