# Peer review of "Methylcellulose–Cellulose Nanocrystal Composites for Optomechanically Tunable Hydrogels and Fibers"

_materials, 2021, doi:10.3390/ma14185137_

Round 1

Reviewer 1 Report

The presented manuscript review and discuss optical and mechanical properties of methylcellulose and nanocomposite materials based on this substance. The review-manuscript described recent findings in the development of cellulose-based materials with “tunable” optical and mechanical properties. In the manuscript, the authors discuss many interesting features related to optical properties of methylcellulose, major of which concerns transparency and scattering properties at different conditions including the addition of nanodopants and temperature variation. The authors provide a comprehensive review of literature related to optical and optomechanical properties of cellulose and its derivatives. The manuscript is well written and in my opinion comprehensive enough to be published in the journal “Materials”. However, I would like to make a few comments, related to the manuscript. 
- The data presented in the manuscript clearly demonstrate that at high temperatures MC solutions turns into turbid gel and its scattering significantly increases. I wonder if you have any data demonstrating temperature-dependent scattering coefficient. Can scattering coefficient of such a MC solution can be adjusted precisely by variation of temperature, or does rapid transition from completely transparent to completely opaque type takes place? How fast is transition from one to another state? If scattering can be precisely controlled by adjusting the temperature of the solution, I believe it could be advantageous for light-controlling devices. 
- I wonder if you are able to find any numerical values related to scattering coefficient of MC solutions? Although some numerical data about light attenuation is presented in the 6 section, I think it would be very helpful for those readers from optical physics to see the approximate value of extinction coefficient at high-temperature conditions. Although this substance is transparent in its liquid low-temperature form I think it would be fruitful to discuss its absorption spectrum in visible or near-infrared region if you have any data on this topic.
- Although a couple of sentences in the Introduction section briefly describes content of the manuscript I would suggest to expand this description and to mention the manuscript content in more details, including references to corresponding sections/subsections if possible.  
Best regards,
Reviewer

Author Response

Reviewer 1

  1. The presented manuscript review and discuss optical and mechanical properties of methylcellulose and nanocomposite materials based on this substance. The review-manuscript described recent findings in the development of cellulose-based materials with “tunable” optical and mechanical properties. In the manuscript, the authors discuss many interesting features related to optical properties of methylcellulose, major of which concerns transparency and scattering properties at different conditions including the addition of nanodopants and temperature variation. The authors provide a comprehensive review of literature related to optical and optomechanical properties of cellulose and its derivatives. The manuscript is well written and in my opinion comprehensive enough to be published in the journal “Materials”. However, I would like to make a few comments, related to the manuscript. 

Response: We thank the reviewer for a careful and systematic evaluation and for providing constructive feedback on our manuscript

  1. The data presented in the manuscript clearly demonstrate that at high temperatures MC solutions turns into turbid gel and its scattering significantly increases. I wonder if you have any data demonstrating temperature-dependent scattering coefficient. Can scattering coefficient of such a MC solution can be adjusted precisely by variation of temperature, or does rapid transition from completely transparent to completely opaque type takes place? How fast is transition from one to another state? If scattering can be precisely controlled by adjusting the temperature of the solution, I believe it could be advantageous for light-controlling devices. 

Response: Upon exceeding the critical temperature (ca. 36-40 °C), the methylcellulose chains undergo LCST transition from random chains to coiled form. The transition is relatively rapid, for example, in the case of a typical gel sample (V = ~5 mL) the transition from a clear dispersion to fully opaque gel takes some tens of seconds after placing the gel vial into a hot water bath above the LCST temperature (of course depending on the vial shape and size, and the surrounding temperature). There the heat convection through the sample material appears as the limiting factor for the speed of transition. In addition, it has been observed that under a mechanical impact, an aqueous methylcellulose sample can form a gel within a period of under 400 μs due to the resulting shockwaves and localized heat generation, which nicely demonstrates the potential rapidity of the transition.[1] Upon cooling down, hysteresis and a somewhat slower transition from turbid to clear is typically observed due to slower re-hydration of the methylcellulose polymer chains when unraveling the coils. The re-hydration efficacy is tunable via dopants, however.[2] In summary, the increase in scattering in methylcellulose gels upon heating is practically instant and stepwise once the critical temperature has been exceeded and driven by the conformational shape change of the polymer chains. Precise gradual adjustments of the scattering behavior do not appear reasonable. Of course, simple on-off switches allowing or preventing light propagation, respectively, could be considered plausible. Perhaps, due to this abrupt and stepwise transition behavior, there does not exist literature or explicit data focusing on the scattering coefficients of methylcellulose, at least according to our knowledge. The interests so far have focused and been mostly driven by mechanical behavior and molecular conformational changes explaining that behavior.

  1. I wonder if you are able to find any numerical values related to scattering coefficient of MC solutions? Although some numerical data about light attenuation is presented in the 6 section, I think it would be very helpful for those readers from optical physics to see the approximate value of extinction coefficient at high-temperature conditions. Although this substance is transparent in its liquid low-temperature form I think it would be fruitful to discuss its absorption spectrum in visible or near-infrared region if you have any data on this topic.

Response: To the best of our knowledge there is no literature specifically exploring the light scattering behavior of MC solutions or gels at high temperature conditions. The interest has been limited in and reported mostly from dried solid films that have been prepared at lower temperature conditions, thus, taking advantage of material transparency at such conditions. However, some optical parameters of solid MC films have been reported, for example, by Atta et al. (2021, preliminary preprint).[3]

  1. Although a couple of sentences in the Introduction section briefly describes content of the manuscript I would suggest to expand this description and to mention the manuscript content in more details, including references to corresponding sections/subsections if possible.  

Response: The contents of the review are now described in more detail in the introductory chapter. The manuscript text has been modified accordingly.

Reviewer 2 Report

The review is devoted to the mechanical and optical properties of methylcellulose containing one-dimensional cellulose nanocrystals. The nanocomposite hydrogels based on the abovementioned materials are important because of their wide application in different areas.

The paper is well organized and can be published in the present form.

Author Response

Reviewer 2

The review is devoted to the mechanical and optical properties of methylcellulose containing one-dimensional cellulose nanocrystals. The nanocomposite hydrogels based on the abovementioned materials are important because of their wide application in different areas.

The paper is well organized and can be published in the present form.

Response: We thank the reviewer for carefully evaluating our manuscript and providing constructive feedback.

Reviewer 3 Report

This review focuses on the recent de- 15velopment in optically and mechanically tunable hydrogels derived from methylcellulose and 16methylcellulose-cellulose nanocrystal composites. I have some comments here.

  1. The titles of drawing need to be simplified. The source and explanation of the subgraph shoud be  be described in the text.
  2. Some sentences are too lengthy and can be more concise.
  3. The increase or decrease of properties and parameters mentioned in this paper can be expressed quantitatively.
  4. Temperature conditions are mentioned many times in this paper, such as ambient temperature, moderate heating, etc.  please give the specific scope.
  5. The preparation of composite hydrogels and how to extract highly ductile and strong fibers can be more specific.

Author Response

Reviewer 3

This review focuses on the recent development in optically and mechanically tunable hydrogels derived from methylcellulose and methylcellulose-cellulose nanocrystal composites. I have some comments here.

  1. The titles of drawing need to be simplified. The source and explanation of the subgraph shoud be  be described in the text.

Response: We assume that the reviewer meant the figure captions and figure panels. However, we have added some clarifications to the figure caption and text.

  1. Some sentences are too lengthy and can be more concise.

Response: In the revised version of the manuscript, we have made changes wherever necessary.

  1. The increase or decrease of properties and parameters mentioned in this paper can be expressed quantitatively.

Response: Numbers have been provided in the manuscript where it has been considered important for understanding the context, benefits and limitations of the materials in question. We concluded it would make the text too long and too heavy for the readers if each value of each particular composition discussed in the text had been written out plainly as numbers. Instead, the trends and magnitudes often better convey the concepts and relative significances and, thus, were preferred. Of course, without forgetting important anchoring values for reference. For further detailed numerical information, the readers are encouraged to read the original publications which are referred to in the manuscript. However, some numerical values have been added to certain parts of the manuscript where considered to clearly add more value to the reader.

  1. Temperature conditions are mentioned many times in this paper, such as ambient temperature, moderate heating, etc.  please give the specific scope.

Response: The concepts of ambient temperature and room temperature mentioned in the manuscript, refer to the, ca. 22 °C (by definition ambient temperature is between 22-25 oC). It also indicates that the temperature has been taken as it is, and it has not been specifically modified, nor has there been an explicit need to control it during the experiments. The numerical clarification of those concepts has been added to the manuscript in appropriate places. The term “moderate heating” mentioned once in the text, referred to the 37 °C in the preceding sentence before it. The connection and the meaning of the “moderate heating” in that specific part of the text has now been made more obvious.

  1. The preparation of composite hydrogels and how to extract highly ductile and strong fibers can be more specific.

Response: More details have been added to the sections describing the gel preparation and fiber spinning processes in the manuscript.

Reviewer 4 Report

This article is informative, clear and well written, focused on the methylcellulose related materials for preparing optically and mechanically tunable hydrogels. I think quality is sufficient to satisfy the requirements of publication.

Author Response

Response: We thank the reviewer for carefully evaluating our manuscript and providing constructive feedback.

Round 2

Reviewer 3 Report

No more comments.